# Incorporating Cascade Effects of Genetic Testing in Economic Evaluation: A Scoping Review of Methodological Challenges

**DOI:** 10.3390/children8050346

**Published:** 2021-04-27

**Authors:** Alexandra Cernat, Robin Z. Hayeems, Lisa A. Prosser, Wendy J. Ungar

**Affiliations:** 1Child Health Evaluative Sciences, Hospital for Sick Children, Toronto, ON M5G 0A4, Canada; cernata@mcmaster.ca (A.C.); robin.hayeems@sickkids.ca (R.Z.H.); 2Institute of Health Policy, Management and Evaluation, University of Toronto, Toronto, ON M5T 1P8, Canada; 3Susan B. Meister Child Health Evaluation and Research Center, Department of Pediatrics, University of Michigan Medical School, Ann Arbor, MI 48109, USA; lisapros@med.umich.edu; 4Department of Health Management and Policy, University of Michigan School of Public Health, Ann Arbor, MI 48109, USA

**Keywords:** cascade testing, genetic testing, economic evaluation, methodology, scoping review

## Abstract

Cascade genetic testing is indicated for family members of individuals testing positive on a genetic test, and is particularly relevant for child health because of their vulnerability and the long-term health and economic implications. Cascade testing has patient- and health system-level implications; however cascade costs and health effects are not routinely considered in economic evaluation. The methodological challenges associated with incorporating cascade effects in economic evaluation require examination. The purpose of this scoping review was to identify published economic evaluations that considered cascade genetic testing. Citation databases were searched for English-language economic evaluations reporting on cascade genetic testing. Nineteen publications were included. In four, genetic testing was used to identify new index patients—cascade effects were also considered; thirteen assessed cascade genetic testing strategies for the identification of at-risk relatives; and two calculated the costs of cascade genetic testing as a secondary objective. Methodological challenges associated with incorporating cascade effects in economic evaluation are related to study design, costing, measurement and valuation of health outcomes, and modeling. As health economic studies may currently be underestimating both the cost and health benefits attributable to genetic technologies through omission of cascade effects, development of methods to address these difficulties is required.

## 1. Introduction

Genetic testing is a powerful diagnostic tool that can be applied to diseases affecting any organ system in the body [1,2,3,4]. These diseases may be hereditary, or mutations can arise de novo. Identification of a genetic diagnosis in a patient facilitates more appropriate management and enables physicians to better approximate patients’ prognoses. Importantly, identifying a genetic diagnosis in a patient enables genetic testing or clinical screening of their family members to determine whether they may also be at risk of developing the disease [3,5]. The first person in a family to undergo genetic testing is known as the proband, index case, or index patient, and the process of subsequently testing or screening relatives is called cascade testing or cascade screening. Cascade health service use has consequences for patients’ relatives, as well as for the health system as a whole. For example, cascade testing may lead to initiation or cessation of periodic screening and surveillance, uncover the need for prevention measures (e.g., implantation of an implantable cardiac defibrillator to ameliorate the risk of sudden cardiac death in the case of cardiomyopathy (CMP)), trigger reproductive decision making, or trigger lifestyle modifications in family members [6]. There are also costs to the health care system: physician and genetic counseling fees for pre- and post-test counseling and follow-up appointments, the cost of medical supplies and technician time for testing and screening, and potentially some pharmaceutical costs [7]. Cascade health service use is particularly relevant for child health because children can both trigger health service cascades and be included in them. When genetic or genomic testing is performed in the pediatric setting, the patient’s parents or siblings may undergo genetic testing or clinical screening as well to determine whether they are at-risk of disease or to help establish disease etiology.

Economic evaluations have been conducted to assess the cost-effectiveness of genetic testing in various patient populations [8,9]. A past review examined cost-effectiveness analyses (CEAs) for Lynch syndrome that incorporated cascade effects and highlighted how changing the number of relatives included in an analysis could modify the results of an economic evaluation [10]. However, methods for economic evaluation stipulated by health technology assessment (HTA) agencies in Canada, the United States, and the United Kingdom fail to account for the cascade effects of a new technology [11,12,13]. As genetic and genomic technologies are implemented more widely, it is increasingly important to consider how cascade consequences ought to be included, since genetic testing performed in a patient enables surveillance in family members as part of recommended clinical management. While the cascade health service consumption triggered in family members can increase health system costs, it can also improve quality and length of life in family members through risk mitigation, preventative care, or earlier diagnosis of a rare condition where symptoms are not yet presenting [6,14]. However, there are methodological challenges associated with incorporating cascade health service use in economic evaluation of technologies directed at children and adults. Specifically, inclusion of health service use by individuals other than the index patient challenges how economic evaluations are designed—and how costs and health outcomes are measured, valued, and modeled. These difficulties are exacerbated when both children and adults are considered in an economic evaluation simultaneously. While there has been a growing literature on spillover effects related to valuing the costs and quality of life of caregivers and family members of patients [15,16,17,18], the challenges associated with incorporating cascade health service use as an integral component of patient care have not been adequately described. Understanding these issues, particularly with regard to child health, is a critical first step toward development of the needed methods.

This scoping review presents a compilation of economic evaluations published to-date that considered the cascade costs and effects of genetic testing in children and adults. Studies were conducted with one of two aims: either to assess the overall impact of implementing a genetic technology for index case identification, or to evaluate only the cascade that ensues. The purpose of the current review was to provide an overview of the literature and to advance economic evaluation methods by identifying methodological challenges and potential solutions for incorporation of cascade effects, with an emphasis on child health. Given these aims, a scoping—rather than systematic—review was most appropriate [19,20]. The guiding research question was: How can methods for economic evaluation be modified to include cascade effects from genetic testing to improve the comprehensiveness and quality of evidence?

## 2. Materials and Methods

A scoping review was conducted in accordance with the methodological framework laid out by Arksey and O’Malley [19]. The review was reported using the Preferred Reporting Items for Systematic Reviews and Meta-Analyses (PRISMA) statement and checklist on scoping reviews [21]. A protocol for this review was not registered in advance of the search.

### 2.1. Search Strategy

Ovid Medline and Embase were searched from database inception to 5 January 2021 for economic evaluations that examined cascade consequences of genetic testing. The search strategy (Appendix A) combined terms describing: economic evaluations and decision analytic modeling; cascade testing, family, and burden; and genetic testing. Search terms included: economic evaluation, cost-benefit analysis, cost-effectiveness analysis, cost-utility analysis, or economic models; cascade, family, mother, father, sibling, cost, or consequence; and genetic testing, genetic predisposition to disease, cascade testing, carrier screening, or variant analysis. Terms relating to spillover effects were also included in the search. Health spillovers refer to the impact of a patient’s illness and treatment on caregivers and non-caregiving family members but are distinct from cascade effects because people who experience spillover effects in the absence of cascade testing or screening do not themselves undergo any testing or screening as part of the patient care plan. However, some of the literature may be relevant, because incorporation of health spillovers in economic evaluation also requires consideration of individuals other than the patient [18]. The electronic search was supplemented with a manual search consisting of a review of the reference lists of included papers. Publications known to us that were not captured in the search were also included.

### 2.2. Eligibility

Publications eligible for inclusion were English-language economic evaluations that considered the cascade or spillover costs and health effects of any genetic or genomic technologies. Studies may have been assessing the technologies in terms of index patient identification or diagnosis, or they may have been evaluating the consequences of implementing cascade testing or screening strategies to identify potentially affected relatives of index cases. Although cost analyses are not full economic evaluations, they were eligible for inclusion for the purposes of this review. The search was not limited by disease state, and economic evaluations or cost analyses where genetic testing was performed in prenatal care were eligible. The search was also not limited based on study location. Only peer-reviewed, published studies were eligible. Ineligible studies were non-English, animal or in vitro, or qualitative; as well as theses, case reports, case series, editorials, commentaries, and conference abstracts were ineligible. Studies that did not present primary data (i.e., reviews) were excluded. Clinical practice guidelines were excluded. Studies in which probands were not diagnosed through genetic testing or the method of proband diagnosis was unclear were excluded, as were studies that did not examine cascade or spillover effects.

Titles and abstracts were reviewed by a single reviewer and full-text articles were obtained for those studies meeting the eligibility criteria. Full-text articles were also obtained to establish eligibility in cases where the title and abstract alone were insufficient. A PRISMA flow diagram was constructed to depict the flow of information through the review.

### 2.3. Data Charting and Analysis

For all included studies, one reviewer independently collected data using a data charting form. Information recorded included: bibliographic information, study purpose, study design, participant characteristics, main findings, and study strengths and limitations relating to consideration of cascade or spillover effects as identified by the paper’s authors. The study design and methods of included papers were then summarized to lay a foundation for presentation of the methodological challenges associated with incorporating cascade effects in economic evaluation.

Included studies were not subjected to critical appraisal, since the objective was to report findings and approaches in the absence of guidelines for incorporating cascade effects. The goals of this review were to report on all available literature as well as to discuss methodological challenges.

## 3. Results

The literature search yielded 357 references after removal of duplicates. Ninety-three studies proceeded to full-text review. Four additional articles not captured in the electronic search were also included; these consisted of one retrieved through manual search of the reference lists of included papers, and three that were previously known. In total, 19 studies were included (Figure 1), all of which addressed cascade effects rather than spillover effects [7,22,23,24,25,26,27,28,29,30,31,32,33,34,35,36,37,38,39]. The eligible papers were published between 1999 and 2020 [22,25,28]. Most publications studied genetic testing for familial hypercholesterolaemia (FH) [22,23,27,30,31,33,34,35] or inherited heart diseases such as CMP [7,24,29,36,39]. Studies were set in a variety of locations: Australia [7,22,23,29,35,38], Canada [25,26], the United States [24,32], and across Europe [27,28,30,31,33,34,36,37,39].

Fifteen of the included studies assessed both costs and health outcomes [7,22,23,26,27,28,29,30,31,32,33,34,36,37,39]. Fourteen of these were CEAs or cost-utility analyses (CUAs) [7,22,23,27,28,29,30,31,32,33,34,36,37,39], while one was a cost-minimization analysis (CMA) [26]. The other four publications only reported on costs [24,25,35,38]. Although Stark and colleagues [38] conducted a CEA and CUA, they only accounted for costs when addressing cascade effects. In general, studies were designed to assess cascade genetic testing or screening strategies for the identification of at-risk relatives of probands [7,22,23,25,26,29,30,31,33,34,35,36,39] rather than to analyze the consequences of using genetic testing to identify or diagnose index patients [27,28,32,37]. A total of three studies focused on children as the proband [24,32,38], while five studies included children as relatives who underwent cascade testing or screening [22,25,26,31,35]. Nine of the included studies adopted a lifetime time horizon [7,22,27,29,30,33,34,37,39].

In studies where decision analytic trees or Markov models were constructed, decision pathways and health states typically reflected the clinical course of family members rather than probands, or the clinical courses for probands were reflected in separate models [7,22,23,25,26,29,30,33,36,39]. The probability of identifying a causative mutation in a proband still appeared within the decision tree, but only insofar as it could inform relatives’ trajectories [7,25,26,29,32,39]. Despite the effort to keep index patients and their relatives separate, many analyses considered the cost of cascade testing and screening to include the cost of the initial genetic test in the proband [7,22,23,25,26,27,29,31,32,33,34,36,37,39]. In contrast, health effects did not receive the same treatment, with outcomes being defined as changes in the quality or quantity of life years of relatives only [7,22,23,29,30,32,33,39]. When children were included in the decision analytic model, they were either included as the sole population [22], or they were modeled together with adults [25,26,31]. When the latter occurred, the decision tree was typically structured such that adults and children had different branches, defined by different parameter values.

### 3.1. Pediatric-Focused Studies

Eight studies [22,24,25,26,31,32,35,38] focused on children either as a primary or ancillary group. Ademi and colleagues [22] examined the cost-effectiveness of cascade screening of children for FH compared with no screening. They constructed a decision-analytic Markov model that considered a cohort of 1000 hypothetical ten-year-old children suspected of having heterozygous FH based on the presence of the disease in one or more of their first-degree relatives (i.e., the modeled children were not the probands). Children who underwent genetic testing for FH and who began statin treatment after receiving a genetic diagnosis were compared to children who did not undergo genetic testing, but who initiated statin treatment either when they reached 25 years of age, or earlier if they experienced a cardiovascular event. The model adopted a lifetime time horizon and evaluation was undertaken from the Australian public health system perspective. Costs, life years gained (LYGs), and quality-adjusted life years (QALYs) for the included children were calculated and reported.

Alfares et al. [24] reported the results of genetic testing in 2912 probands with hypertrophic CMP and 1209 of their asymptomatic relatives. Of the included probands, 462 (16%) were pediatric (aged 16 years or younger), 2412 (83%) were adults, and age was not provided for the remaining 38 (1%). The ages of the asymptomatic family members captured in analysis were not reported. Understanding the health system consequences of cascade genetic testing was a secondary aim to presenting the clinical genetics data. Only a brief cost analysis was conducted, reporting the total lifetime cost of periodic screening avoided for those relatives who received a negative genetic testing result.

Bapat and colleagues [25] conducted a cost comparison of predictive DNA testing versus conventional clinical screening for individuals who have a family history of familial adenomatous polyposis (FAP) from the third-party payer perspective. Both pediatric and adult relatives were considered, with onset of clinical screening occurring at age 15 years for 60% of relatives, at 25 years for 20% of relatives, and at 35 years for 20% of relatives. The costs calculated and reported included the cost of genetic testing in probands, the cost of genetic testing of relatives in the predictive DNA testing strategy, and the cost of routine clinical screening (flexible sigmoidoscopy) in family members.

Chikhaoui et al. [26] conducted a CMA comparing the direct costs of predictive genetic testing and conventional clinical screening in the first-degree relatives of FAP patients. A decision analytic Markov model spanning a 40-year time horizon was constructed, with relatives entering the model at 12 years of age. Clinical screening costs were only applied for family members, while genetic testing costs were applied for both the proband and their relatives.

Li and colleagues [32] examined the cost-effectiveness of alternative genetic testing strategies to diagnose unexplained global developmental delay and intellectual disability in children. A decision analytic tree was constructed to compare the following strategies: chromosomal microarray analysis of index cases only; chromosomal microarray analysis of index cases followed by parental chromosomal microarray analysis if a variant of unknown significance (VUS) was identified; or chromosomal microarray of index cases followed by either parental chromosomal microarray analysis if a VUS was identified, or targeted next generation sequencing (NGS) of index cases with a VUS but whose parents were unavailable for testing, as well as targeted NGS of index cases whose chromosomal microarray analysis was negative. The modeled study cohort consisted of 1000 index cases. The time horizon for analysis was one year and the perspective adopted was that of the United States health care payer. Costs and number of genetic diagnoses were reported for each strategy. This paper differed from others included in this review in that cascade genetic testing was not conducted for the benefit of the probands’ relatives, but rather to determine inheritance of a variant identified in the index case. However, studying the origin of a variant—and whether it is de novo or inherited—is routine in medical genetics and the costs of these confirmatory genetic tests are therefore relevant for economic evaluation of genetic testing technologies directed at probands. Although parents may be asymptomatic in these investigations, a positive finding may precipitate ongoing surveillance. It is for these reasons that this paper was included.

Pang et al. [35] evaluated the clinical outcome of parent-child cascade genetic testing for FH. As a secondary objective, they estimated the additional cost of treating identified cases of FH through childhood. From 126 adult parents known to have a causative FH mutation with children aged 18 years or younger, 244 children and adolescents were identified for screening. Of these, 148 (61%) underwent genetic testing. Costs of statin therapy were calculated for those children that received a positive genetic testing result, from the time treatment was commenced until the child reached age 18 years. No other costs (for example, the cost of genetic testing) were taken into account.

Finally, Stark and colleagues [38] investigated the clinical outcomes and cost-effectiveness of implementing genomic sequencing to diagnose infants suspected with rare monogenic disorders. They estimated the cost of cascade genetic testing in the infants’ parents; however, this was a minor component of their study and the cost calculation methods were not provided.

Finally, Lazaro et al. [31] modeled probands and their family members together. Their study is discussed in the following section.

### 3.2. Family-Focused Studies

There were four studies in which probands and their family members were considered simultaneously [27,31,34,37]. Crosland et al. [27] compared the cost-effectiveness of nine different strategies for identifying individuals with FH, taking cascade effects into account. The model population consisted of current or known FH index cases, potential new index cases, and the relatives of people in both of those groups. In other words, probands and relatives were modeled together. In the decision analytic tree, there was one arm for each of the nine strategies. The first chance node in each arm indicated a type of individual: those with a clinical diagnosis of definite FH; relatives of individuals known to have monogenic FH; people identified as new index cases through a primary care database search; relatives of newly identified index cases; people with an early myocardial infarction (MI); and relatives of those with early MI. The proportions or probabilities of falling into each of those six categories were determined beforehand based on published literature. A subset of index cases, both previously known and newly identified, underwent genetic testing to enable cascade genetic testing of their family members. The health outcomes of interest included the number and proportion of people with FH who were diagnosed versus undiagnosed, as well as QALYs. The tree was folded back to determine the expected values for lifetime costs and QALYs of diagnosing FH. Costs and QALYs were considered for the entire cohort without distinguishing between index patients and relatives.

Lazaro and colleagues [31] assessed the cost-effectiveness and cost-utility of a family-based national genetic screening program for FH, compared with the lack of such a program, in Spain. They modeled a hypothetical cohort, of which one-third was index cases and two-thirds were family members. These proportions were predetermined based on the ratio observed in the Spanish Familial Hypercholesterolemia Cohort Study registry [40]. All subjects were affected by FH, but the decision in the decision tree was whether they would be treated as such. Subjects in the intervention arm of the decision tree were genetically diagnosed with FH, received appropriate management, and had a lower probability of experiencing a coronary event [31]. Individuals in the usual care arm also had FH but were not genetically diagnosed. They were therefore treated as having high cholesterol and had a greater chance of a coronary event over a period of ten years following the implementation of the screening program. Modeled family members were adults and children aged three years or older. Although both index patients and their relatives were included in the same model, the intervention and no intervention arms of the decision tree were divided such that probands and family members followed different pathways, with different probabilities of experiencing a coronary event or death. The decision tree branches pertaining to relatives were further divided so as to separate adult family members and children, and allow for their trajectories through the model to be constrained by different parameter values. Health state utilities for adults and children were different, with children having a higher utility for their baseline health state than adults but experiencing greater deterioration in quality of life following a coronary event. Baseline health state utilities were obtained from the Spanish National Statistics Unit and were used to calculate QALYs.

Nherera et al. [34] evaluated the cost-effectiveness of four alternative cascade screening strategies for FH: (i) use of elevated LDL-C to identify affected relatives of FH patients; (ii) DNA testing to identify an FH-causing mutation in an index patient, followed by cascade genetic testing in relatives; (iii) DNA testing to identify an FH-causing mutation in an index patient, cascade genetic testing in relatives, and cascade clinical screening using LDL-C levels in the relatives of definite FH index cases who are mutation-negative; and (iv) DNA testing to identify an FH-causing mutation in the index patient, cascade genetic testing in relatives, and cascade clinical screening using LDL-C levels in the relatives of definite and probable FH index cases who are mutation-negative. The study modeled a hypothetical cohort of 1000 suspected FH index patients and their relatives. First, the authors constructed a decision tree to determine how many index cases would be identified as true/false positives for FH and true/false negatives using each of the four strategies listed above. The relatives of true positive probands in all four strategies, the relatives of false negative probands in the third strategy, and the relatives of false positive probands in the first and fourth strategies were offered cascade testing. The number of family members tested per proband and the number identified as true/false positives and true/false negatives was determined based on published literature. All index patients and relatives then entered a Markov model for a statin treatment protocol, together. The researchers first reported total costs and total QALYs separately for the groups of index patients and family members. They then determined mean cost and mean QALY per person for the combined patient-relative cohort to compare the incremental cost-effectiveness of alternative strategies over a lifetime time horizon.

Lastly, Sie and colleagues [37] examined genetic testing for identification of Lynch syndrome patients in individuals presenting with colorectal cancer (CRC) at age 70 years or below—compared to testing patients age 50 years or below. They were interested in detection of Lynch syndrome in CRC patients and their family members. Rather than attempting to combine index cases and relatives in one decision analytic model, Sie et al. developed three models. The first was focused on comparing the efficacy of testing in index patients 70 years or younger versus testing in index patients aged 50 years or younger. The second model evaluated cost-effectiveness of testing CRC patients aged 51–70 years compared with not testing them. Costs and life-years were determined for each group. Family members were not included. The final model focused on the cost-effectiveness of genetic testing in the relatives of CRC patients aged 51–70 years who were identified as having Lynch syndrome compared to genetic testing in the relatives of CRC patients aged 50 years or below. Index patients were not included in this model. The researchers subsequently summed the results for index patients with those of their relatives (i.e., findings from models two and three) to determine the incremental cost per life year gained when patients aged 51–70 years underwent genetic testing compared to testing in those who were aged 50 years or below. Detailed information about each included study can be found in Table 1.

## 4. Discussion

As genetic and genomic technologies are implemented with increasing frequency in the clinical setting, incorporating cascade effects in economic evaluation has become increasingly important. Cascade genetic testing and screening may lead to earlier detection and treatment of a condition in family members. This, in turn, may result in improved health outcomes, whether in the form of increased life years, improved quality of life, or combined as QALYs. Omission of cascade effects leads not only to an underestimation of the costs associated with a health technology, but also of the benefits. Moreover, there seems to be a discord in which a technology is valued for its potential to catalyze better management of a health condition, but in which cascade effects are not evaluated when systematically assessing the technology to inform a funding or policy decision.

This scoping review provided an overview of the economic evaluations conducted to-date that have considered cascade effects of genetic testing in adults and children, and revealed two main foci: evaluating cascade testing or screening strategies for identification of at-risk or affected family members [7,22,23,25,26,29,30,31,33,34,35,36,39]; or assessing the costs and consequences of using alternative genetic testing technologies for identification or diagnosis of index patients [27,28,32,37]. Guidelines for economic evaluations focused on the costs and benefits of patients by definition [11]. Despite these guidelines, some economic evaluations focused on the relatives of index cases because they recognized that cascade costs and health effects can be substantial and have non-trivial implications for patient care and health systems decision making.

### 4.1. Challenges to Incorporation of Cascade Effects in Economic Evaluations

Based on close examination of included studies, there are a number of methodological challenges common to economic evaluations of cascade effects; they are related to study design, costing, and measurement and valuation of health outcomes (Figure 2). These are summarized in Table 2, along with alternate suggested approaches.

#### 4.1.1. Study Design

An intervention in an index case may lead to multiple cascades. There is the primary cascade, where first-degree members of the index patient’s family (i.e., parents, siblings, or children) undergo testing or screening to assess their risk of developing a particular disease. This is the cascade health service use that the studies included in this review sought to evaluate. However, when considering family members who, at the time of their testing or screening, have not yet reached reproductive age or have not yet had children, it is possible to foresee a secondary cascade, whereby these relatives’ future pregnancies and/or future children will receive screening as well. One of the decisions that must be made in the study design phase is whether this secondary cascade should be included. This would have a number of implications, from the selection of an appropriate time horizon, to the types of health outcomes considered. Models that extend over future generations are limited by the inability to predict future changes in medical practice or the advent of new technology. One option to address this difficulty may be to model the current generation as a primary analysis and add secondary cascades in future generations in a scenario analysis, highlighting the uncertainties in model construction.

Another challenge to study design is the choice of time horizon in the reference case. Economic evaluation guidelines state that the most appropriate choice of time horizon is the individual’s lifetime [41], but this approach may be problematic when cascade health services are included because a decision must be made as to which lifetimes are being considered. It is likely that probands and family members will have differing life expectancies, either because some of the included individuals will be children and others will be adults, or because affected individuals may not be expected to live beyond a certain age or a certain number of years following onset. Family members may outlive the proband, or vice versa. The most prudent decision may be to adopt a time horizon based on the lifetime of the youngest individuals included in the study, as this would provide an opportunity to capture as many cascade costs and outcomes as possible.

#### 4.1.2. Cost

Conducting a thorough economic evaluation requires that all health resources consumed within the chosen time horizon be identified, measured, and valued. Understanding the clinical care pathway associated with a disease helps with the identification of relevant resources, but accounting for cascade services adds complexity, especially for genetic conditions with variable penetrance, expressivity, or ages of onset. The surveillance or treatment protocols that need to be initiated in a family member may depend on that person’s age, phenotype, or other risk factors in addition to genetic status. Use of cascade health resources will thus vary widely as a function of an individual’s risk status. There may also be challenges associated with identifying downstream health resource consumption related to treatment of an identified condition. For instance, family members may be prescribed drugs triggered by cascade health resource use.

After health resources are identified, their use must be quantified. Accounting for a range of cascade health resources complicates this task. Different relatives within the same family may require or consume different volumes of resources, or may use resources at different frequencies. Moreover, the volume of resources consumed by an individual may change over time, depending on the progression of disease. It may not always be possible to predict when modifications to a clinical management plan may become necessary. In these situations, resource use measurement may rely on assumptions based on average recommended practice or on input from clinical experts, with uncertainty assessed in sensitivity analyses. This is the approach that was taken by some of the studies included in this review, which used clinical practice guidelines to inform the type and volume of resource use considered [7,26,27,29,34,36,39]. In addition, administrative health insurance databases may also be a valuable tool when conducting economic evaluations that include cascade effects. These databases contain real-world information about a wide range of health resource use within particular health plans or jurisdictions and, if linked with other data sources (such as patients’ electronic medical records), they could help better elucidate patterns of health resource consumption. Such data may be particularly powerful for studying cascade health service use if they allow for linkage of probands’ and family members’ health records.

After resources are identified and quantified, their costs must be determined. For some technologies, the consideration of cascade health service use may complicate this process because it may not always be possible to separate the cost of implementing a technology for index cases from the cost of cascade testing or screening in family members. An example is trio whole genome sequencing [42], in which DNA of a pediatric proband and their parents are processed and sequenced together. Results for primary and secondary genetic variants are reported for children and parents, together with a suitable clinical action plan. Trio costs cannot be divided to obtain the cost of genetic testing separately for the child and parents.

In this review, included studies typically only accounted for the costs associated with cascade testing and screening in family members, as opposed to considering the costs of the proband’s clinical care as well. The cost of the initial genetic test in the index patient was included, however, since that first investigation triggered the cascade of health services. Analysts may wish to consider reporting both disaggregated and aggregated costs to make cost calculations more transparent and allow for a better understanding of what proportion of costs can be attributed to proband genetic testing versus health service use triggered in relatives. When the clinical trajectories of both probands and their family members were modeled, index patients and relatives were treated as one cohort and costs were calculated as they would have been in a traditional economic evaluation [27,31,34]. The exception was the study by Sie and colleagues [37], in which different decision trees were constructed for index patients and family members. In that case, costs pertaining to probands and relatives were calculated separately and then summed together.

#### 4.1.3. Measurement and Valuation of Health Outcomes

The reference case of an economic evaluation should be a CUA, whereby all health outcomes are expressed as QALYs, calculated by multiplying the number of life years an individual spends within a particular health state by a utility that reflects the health-related quality of life in that state [11]. Of particular concern when considering the aggregation of QALYs across family members of different ages is that different instruments and approaches are used to measure utilities in children and adults. Children have less developed cognitive and linguistic abilities than adults, requiring proxy administration. More importantly, the dimensions of health relevant to adults may not be congruent with the dimensions of health relevant to children and adolescents. The relative value of dimensions may also differ across age groups [43,44]. While the methods for obtaining utilities from adults are well established, the methods for doing so in pediatric populations are still being developed. Moreover, results from different tools often cannot be combined into one overall outcome measure. This presents a significant challenge for the inclusion of cascade health effects in economic evaluation, as both adults and children may be referred for cascade investigations—and may therefore experience changes to their quality of life that must be captured in the analysis. Some economic evaluations included in this review addressed this challenge by only considering adult populations [7,23,26,27,28,29,30,34,36,37,39]. Ademi and colleagues [22], on the contrary, considered a cohort of potentially affected relatives composed entirely of children. Even if it were possible to use a single tool to measure health utilities of multiple individuals at different stages of their lives, aggregating health benefits across multiple individuals is problematic, because outcomes such as QALYs are defined and interpreted in terms of an individual’s life expectancy.

The majority of included studies only accounted for QALY gains in immediate family members. However, similar to costing, when the clinical trajectories of both probands and their relatives were modeled, they were considered one cohort and QALYs were calculated as in a traditional economic evaluation. Sie et al. [37] did use separate decision trees for index patients and their relatives and calculated QALYs for each type of individual separately. They then added the results from each group together to determine overall QALY gains in the cohort as a whole. Reporting disaggregated outcomes in addition to any aggregated outcome measures may be important to ameliorate some of the challenges of aggregating QALYs described in the paragraph above.

The use of health utilities to capture health-related spillover effects and the incorporation of these utilities in CEAs has been previously explored [18]. Although health spillover effects are a distinct concept, some of the methodological challenges they pose are also raised by cascade effects. For example, Wittenberg and colleagues [18] noted that the most common approach of including spillover effects in economic evaluation is currently to sum patient and spillover QALYs. This approach is problematic because it gives spillover QALYs equal weight to patient QALYs, when it may not always be appropriate to do so. Wittenberg et al. pointed out that summation of QALYs in this way may unintentionally shift decisions toward benefiting caregivers or family members over the patients themselves because the cumulative increase in QALYs that multiple caregivers and family members experience when an intervention is implemented may be greater (on average) than the increase in QALYs experienced by an individual patient [18]. Further, the magnitude of caregiver QALYs depends directly on the number and availability of caregivers attending to a patient [45]. These are especially important considerations for child health, as parents are expected to experience quality-of-life spillover effects when their children are ill and multiple parents or informal caregivers are typically involved in the care of a sick child. Considering these challenges for valuation of cascade testing, similar approaches could be considered; to report disaggregated outcomes separately for the individual and for those included in cascade testing, and to consider approaches that use a multiplier to prioritize outcomes for the index case [46,47].

#### 4.1.4. Model Design

Decision analytic models constructed for economic evaluations reflect the alternative clinical care pathways and health states associated with the disease under study and the interventions being assessed [11]. The clinical activities and health states that form the model structure, and the associated health costs and outcomes, reflect the interactions with the health care system experienced by individual patients. Developing a model that includes cascade testing and screening in patients’ families is challenging because cascade health service use is tangential and not directly relevant to the clinical trajectory of the index patient.

If a single model consisting of health states and clinical events relevant to index patients as well as their family members were to be constructed, one challenge would be identifying all of the relevant states or events. The clinical pathway followed by index patients may be different than that followed by their family members, and relatives themselves may have different clinical experiences than one another depending on their ages and comorbidities. As a result, a large number of health states or clinical events may need to be considered, especially as heterogeneity within and between families increases. This can add excessive complexity to a model and can make parameter estimation an especially difficult process. Some health states, or transitions between health states, may be common to both index patients and their family members. This raises yet another challenge: an index patient may have one probability of transitioning between health state A and health state B, while their relative may have a different probability of doing so.

The notion of both index patients and their family members passing through the same model simultaneously is problematic. Index patients and their family members are not independent of one another in terms of the health services they consume. Moreover, index patients must enter the model at the beginning and receive an intervention. Relatives, however, necessarily enter the model after their associated index patient and following the delivery of the intervention. Consequently, a model that can include both index patients and their family members would be required. One example is the approach taken by Nherera et al. [34]. In that study, the decision tree model incorporated Markov states that varied as a function of test result and treatment protocol for patient and relatives. The researchers determined and reported expected costs and QALYs separately for index patients and family members. They then combined the values to estimate mean cost and mean QALY per person for the combined patient-relative cohort to compare alternative screening strategies over a lifetime time horizon. Use of more advanced modeling techniques—such as discrete event microsimulations which track the progress of individual persons through health states—may be another approach to address the difficulties identified above. The approach taken to jointly model the costs and health effects of patients and family members may be associated with structural uncertainty that should be thoroughly tested in sensitivity analysis.

#### 4.1.5. Decision-Making

The incorporation of cascade and spillover effects in economic evaluations may lead to equity concerns, as some illnesses may receive greater attention from decision makers than others [18]. Similar concerns apply to the valuation of cascade effects. With regard to cascade effects of genetic testing, uptake of cascade testing or screening may be greater for pathogenic conditions where genetic risk information has implications for an individual’s reproductive decisions. As the number of relatives who access cascade health services increases, so too do the potential aggregate health benefits that should be captured in analysis. As a result, diseases where cascade testing or screening occur more commonly may be prioritized. In the context of child health, this means that funding decisions may be prioritized for pediatric-onset conditions based on the benefits to people other than the affected child. In turn, investment in screening individuals with serious genetic diseases that arise de novo (i.e., are not inherited) may receive less attention.

### 4.2. Limitations

The findings of this review should be interpreted in the context of certain limitations. First, the literature search focused on identifying economic evaluations and cost analyses but did not explicitly consider papers exploring cascade health effects alone. An additional challenge was developing the search strategy, and it is possible some eligible papers were not identified. There is no MeSH term or Emtree subject heading for cascade testing, and authors use a wide variety of terms to describe this process. For instance, some describe it as cascade screening [23,31,34] while others term it predictive genetic testing [25]. Others are more specific and refer to parent-child genetic testing [35]. These terms were included in the search strategy, but they do not always appear in the title, abstract, or keywords list of included papers [24,28].

## 5. Conclusions

Cascade genetic testing and clinical screening triggered by genetic testing in an index case enables the identification of at-risk relatives and the initiation of surveillance protocols and interventions, which may reduce morbidity and mortality in these individuals. However, cascade effects are not currently regularly incorporated in economic evaluations, so cost-effectiveness analyses comparing genetic testing and screening strategies may be underestimating both the costs and health benefits attributable to the implementation of the technology in a particular population. This is important in child health, as genetic testing in a parent may have health implications for their children. In addition, genetic technologies are being used for an increasingly wide range of pediatric indications, with children themselves triggering health service cascades. This scoping review provided an overview of the studies conducted to-date that have attempted to include cascade effects and suggestions for addressing some of the methodological challenges that arise related to measurement of costs, measurement of outcomes, study design and modeling. Development and validation of formal methods by health economists and health services researchers to address these challenges is warranted. Further, in future updates of their guidelines, HTA agencies around the world must explicitly consider cascade effects and optimal approaches for incorporation in economic evaluation.

## Figures and Tables

**Figure 1 children-08-00346-f001:**
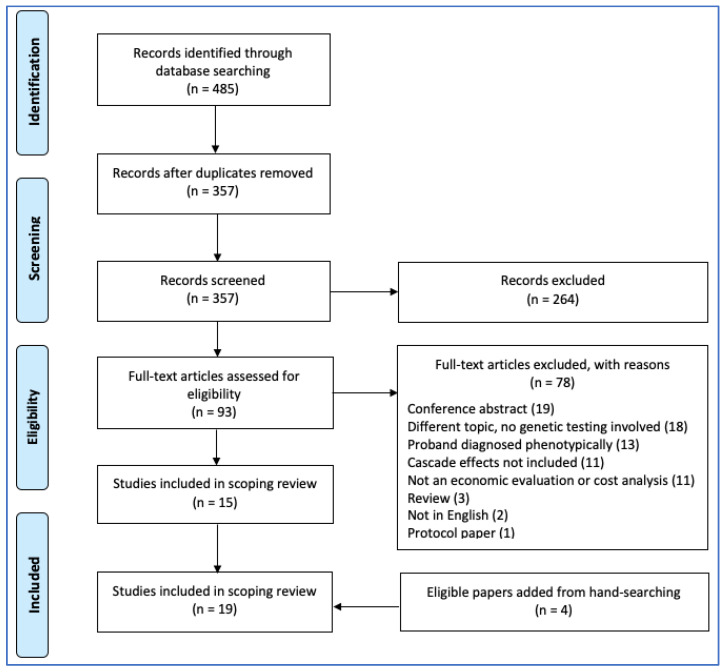
PRISMA flow diagram.

**Figure 2 children-08-00346-f002:**
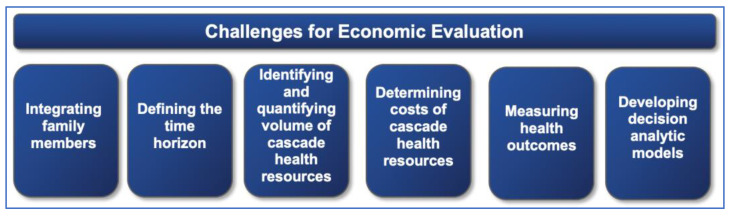
Challenges to economic evaluation methodology associated with incorporation of cascade effects in analysis.

**Table 1 children-08-00346-t001:** Economic evaluations and cost analyses considering cascade costs and health effects of genetic testing.

Author (Year)	Country	Disease	Type of Study	Perspective and Time Horizon	Stated Aims	Strategies Compared	Participants or Modeled Cohort	Measurements
Ademi et al. (2020) [22]	Australia	FH	CEA and CUA	Public health care system perspective. Lifetime time horizon (model ran until participants died or reached age 100 years).	To evaluate the cost-effectiveness of cascade screening of children for FH.	1) Individuals screened and genetically diagnosed with FH; affected individuals treated with statins. 2) Individuals not screened and not treated until clinically identified with FH.	1000 hypothetical ten-year-old children suspected of having heterogeneous FH (i.e., no probands).	Costs, life years (LYs), and QALYs for family members only. Cost of genetic testing in the index patient was included.
Ademi et al. (2014) [23]	Australia	FH	CEA and CUA	Public health care system perspective. Time horizon of 10 years following onset of coronary heart disease.	To evaluate the cost-effectiveness of cascade genetic testing for FH.	1) Relatives of index cases undergo cascade genetic testing supplemented with measurement of LDL-C, followed by statin treatment. 2) Relatives of index cases do not undergo screening or receive follow-up care.	Adult first- and second-degree relatives of genetically confirmed FH index cases (i.e., no probands).	Costs, years of life lived, and QALYs for family members only. Cost of genetic testing in the index patient was included.
Alfares et al. (2015) [24]	United States	Hypertrophic CMP	Cost analysis	Public payer perspective (Medicare). No time horizon.	To describe genetic testing results of a cohort of unrelated probands and their family members; consideration of costs was secondary.	1) Cost of genetic testing costs and estimated clinical screening costs when genetic testing is used to determine risk status in unaffected relatives. 2) Cost of clinical screening costs when genetic testing is not used to determine risk status in unaffected relatives.	2912 unrelated probands and 1209 family members.	Costs of familial genetic testing and surveillance that would be required for genotype-unknown individuals (costs of proband genetic testing not included).
Bapat et al. (1999) [25]	Canada	Familial adenomatous polyposis (FAP)	Cost comparison	Public payer perspective. Time horizon of 40 years.	To compare the costs of predictive genetic testing and conventional clinical screening for identification of individuals who may inherit FAP.	1) Family members of FAP index cases undergo genetic testing followed by appropriate clinical screening. 2) Family members of FAP index cases undergo clinical screening only.	First-degree relatives of index patients with genetically confirmed FAP (i.e., no probands).	Costs of genetic testing and clinical screening in family members, including cost of proband genetic testing.
Catchpool et al. (2019) [7]	Australia	Dilated CMP and other non-hypertrophic CMPs	CUA	Public health care system perspective. Lifetime time horizon.	To assess cost-effectiveness of performing genetic testing in families with dilated CMP compared with clinical surveillance alone.	1) Relatives undergo cascade genetic testing as well as clinical screening. 2) Relatives undergo cascade clinical screening only.	Clinically unaffected adult first-degree relatives whose index case had a clinical diagnosis of dilated CMP and who underwent exome sequencing (i.e., no probands).	Costs and QALYs for family members only. Cost of exome sequencing in the index patient was included.
Chikhaoui et al. (2002) [26]	Canada	FAP	CMA	Public health care system perspective. Individuals enter model at age 12 years and exit when polyps are identified or at age 50 years if polyps not identified.	To compare direct costs of clinical screening and predictive genetic testing strategies for FAP.	1) FAP patient is diagnosed clinically and potentially affected relatives undergo periodic clinical surveillance. 2) FAP patient is diagnosed clinically, undergoes genetic testing, and potentially affected relatives undergo cascade genetic testing followed by appropriate clinical surveillance.	First-degree relatives of FAP patients. Relatives aged 12–50 years.	Costs of genetic testing and clinical screening in at-risk relatives, including cost of proband genetic testing.
Crosland et al. (2018) [27]	United Kingdom	FH	CUA	Public payer perspective. Lifetime time horizon.	To evaluate the cost-effectiveness of different methods to identify FH index cases, while considering cascade testing.	Nine strategies were compared, all using cascade testing combined with different approaches to identify index cases.	Existing FH index cases, new index cases, and potentially affected relatives.	Costs and QALYs for both index cases and their relatives in a combined manner.
Heimdal et al. (1999) [28]	Norway	Inherited breast cancer	CEA	Public payer perspective. Participants are followed from age 35 years to age 60 years.	To estimate the cost of identifying women with a genetic predisposition to breast cancer, and following them with the intention of treating and curing early cancer.	1) Cancer family clinic strategy whereby women at-risk for inherited breast cancer are offered annual clinical examination. 2) Founder mutation strategy whereby all breast and ovarian cancer patients in Norway are tested for *BRCA1* mutations to identify the number of families with *BRCA1*-related disease. Healthy members of these families are offered genetic counseling and testing.	Women known to be at high risk for familial breast cancer, current breast and ovarian cancer patients who might have *BRCA1*-related disease, and the health family members in newly identified *BRCA1* families.	Average cost per cancer detected and average cost per LY gained were presented separately for each strategy, rather than in an incremental analysis. Costs were measured for family members. In the founder mutation strategy, cost of proband genetic counseling was included, but cost of proband genetic testing was not.
Ingles et al. (2011) [29]	Australia	Hypertrophic CMP	CEA and CUA	Public payer perspective. Lifetime time horizon (individuals tracked through health states until death or age 100 years).	To evaluate the cost-effectiveness of the addition of genetic testing to management of hypertrophic CMP families, compared with clinical screening alone.	1) Hypertrophic CMP family presenting for screening and management has access to genetic testing for the proband. 2) Hypertrophic CMP family presenting for screening and management does not have access to genetic testing for the proband.	Clinically unaffected relatives aged 18 years or older of clinically affected hypertrophic CMP individuals (i.e., no probands).	Costs, LYs, and QALYs for family members only, including cost of proband genetic testing.
Kerr et al. (2017) [30]	United Kingdom	FH	CUA	Public payer perspective. Lifetime time horizon.	To estimate the cost-effectiveness of genetic testing in relatives of monogenic FH patients.	1) Genetic testing of individuals clinically diagnosed with FH, cascade genetic testing in the relatives of monogenic probands, and statin treatment. 2) No genetic testing of probands, cascade testing, or treatment of relatives.	Index cases who receive a diagnosis of monogenic FH and their relatives. All individuals were adults aged 20 years or older.	Costs and QALYs. Index patients receive treatment for FH in both the intervention and non-intervention arms. Outcome of the index case’s genetic test does not affect their treatment. No costs or benefits considered for identification and treatment of index cases, nor for treatment of relatives negative for the familial mutation.
Lazaro et al. (2017) [31]	Spain	FH	CEA and CUA	Public payer perspective and societal perspective. Time horizon of 10 years following implementation of screening program.	To assess the cost-effectiveness of a national genetic cascade testing program for FH in Spain.	1) Implementation of a national genetic testing program whereby genetic testing is used to identify FH index cases and their relatives. 2) No such program is implemented.	9,000 FH patients (2250 index cases and 6750 relatives). One-third of included relatives are children aged 3 years or older.	Costs, coronary events avoided, deaths avoided, and QALYs for all 9000 individuals in a combined manner.
Li et al. (2018) [32]	United States	Unexplained developmental delay or intellectual disability	CEA	Public payer perspective. One-year time horizon.	To compare the cost-effectiveness of several genetic testing strategies for the genetic diagnosis of patients with unexplained developmental delay.	Two decision trees. The second, relevant here, compared: 1) Chromosomal microarray without follow-up in case of variant of unknown significance (VUS) or negative result. 2) Chromosomal microarray followed by parental chromosomal microarray in case of VUS in the child. 3) Same as second strategy, but negative chromosomal microarray in the patient is followed-up with next-generation sequencing.	Cohort of 1000 patients, and some of their parents.	Costs of chromosome microarray in patients and, where applicable, costs of chromosomal microarray in their parents. Effectiveness measure (number of genetic diagnoses) pertained only to the patients.
Marang-van de Mheen et al. (2002) [33]	The Netherlands	FH	CEA	Public payer perspective. Lifetime time horizon, until 85 years of age.	To estimate the cost-effectiveness of the current FH cascade screening program in The Netherlands.	1) Implementation of a national family-based genetic testing program for FH, whereby first- and second-degree relatives of probands diagnosed with FH undergo genetic testing. 2) No such program implemented.	2229 first- and second-degree relatives of 137 genetically diagnosed FH index patients (i.e., no probands considered in the model).	Costs and LYs for relatives only, including cost of proband genetic testing, were presented separately for each strategy rather than in an incremental analysis.
Nherera et al. (2011) [34]	United Kingdom	FH	CUA	Public payer perspective. Lifetime time horizon.	To estimate the cost-effectiveness of four different cascade screening methods for FH.	1) Identification of affected relatives clinically. 2) Genetic testing in index patients and cascade genetic testing in mutation-positive probands to identify affected relatives. 3) Same as the second strategy, but additionally, cascade clinical screening in relatives of mutation-negative index cases who definitely have FH. 4) Same as the third strategy, but additionally, cascade clinical screening in relatives of mutation-negative index cases who probably have FH.	1000 people suspected of heterozygous FH. All modeled individuals were adults.	Costs and QALYs for index patients and relatives separately as well as together.
Pang et al. (2018) [35]	Australia	FH	Cost analysis	Public payer perspective. Costs were counted for children from age 10–18 years.	To evaluate the clinical outcome of cascade genetic testing children of FH patients and (as a secondary aim) to determine the additional cost of treating each child.	No comparators. Identification and treatment costs for children identified as FH patients were calculated.	84 mutation-positive children from 80 affected parents. Of the 84 identified children, 40 began treatment with low-dose statins.	Costs of an individual receiving statins from age 10–18 years. No genetic testing costs were included.
Sabater-Molina et al. (2013) [36]	Spain	Inherited cardiac diseases	CEA	Public health care system perspective. Costs were counted throughout relatives’ lifetimes, from age 10–60 years.	To calculate cost of genetic testing in probands and their relatives and compare those costs with costs of clinical tests avoided in mutation-negative individuals.	1) Identification of genotype-negative family members of index patients and subsequent cessation/avoidance of clinical surveillance. 2) No such identification and cessation.	234 non-related index cases with hypertrophic CMP, arrhythmogenic right ventricular CMP, long QT syndrome, and Brugada syndrome. 738 relatives of these probands, of whom 371 were genotype-negative and included in analysis.	Costs and diagnostic yield for probands and their relatives reported separately. Costs of clinical examination in probands or carrier family members, and costs of genetic testing in carrier relatives were not included.
Sie et al. (2014) [37]	The Netherlands	CRC	CEA	Public health care system perspective. Lifetime time horizon.	To assess the cost-effectiveness of increasing the age limit for genetic testing in CRC patients, including cascade genetic testing.	Three decision models were constructed. The third, most relevant here, compared: 1) Relatives of CRC patients aged 51–70 years identified as having Lynch syndrome undergo cascade genetic testing 2) Relatives do not undergo cascade genetic testing.	112 CRC patients identified as having Lynch syndrome, and 935 relatives.	Costs, number of Lynch syndrome patients identified, and LYs for index patients and relatives separately, and then are added together.
Stark et al. (2019) [38]	Australia	Rare monogenic disorders	Cost analysis	Public payer perspective. No time horizon (one-time cost of cascade genetic test).	To investigate the clinical and health economic impacts of genomic sequencing for rare-disease diagnoses.	No comparators. Costs of genetic testing and counseling in probands’ family members were calculated.	88 first-degree relatives of children with suspected monogenic disorders, of whom 79 underwent cascade genetic testing.	Conducted a CEA and CUA as part of the study, but only costs for genetic testing of probands’ parents were considered with respect to cascade services.
Wordsworth et al. (2010) [39]	United Kingdom	Hypertrophic CMP	CEA	United Kingdom hospital perspective. Lifetime time horizon.	To explore the cost-effectiveness of four different methods of cascade testing and screening.	1) Cascade genetic testing in relatives. 2) Cascade clinical screening in relatives. 3) Cascade clinical screening with five yearly repeat clinical investigations. 4) Cascade genetic testing with five yearly repeat clinical investigations for those whose parents’ DNA mutation was not identified.	Adult asymptomatic children of probands diagnosed with hypertrophic CMP.	Costs and expected years of life for family members only, including cost of proband genetic testing.

CEA: cost-effectiveness analysis; CMA: cost-minimization analysis; CUA: cost-utility analysis; LY: life-year; QALY: quality-adjusted life year; CMP: cardiomyopathy; CRC: colorectal cancer; FAP: familial adenomatous polyposis; FH: familial hypercholesterolemia; LDL-C: low-density lipoprotein cholesterol; VUS: variant of unknown significance.

**Table 2 children-08-00346-t002:** Summary of methodological challenges of incorporating cascade effects in economic evaluation and suggestions for alternate approaches.

Component of Economic Evaluation	Challenges	Alternate Approaches
Study Design	An intervention in an index case may lead to multiple cascades across a widening sphere of relatives and/or more than one generation of a family.	Model the current generation in the nuclear family as a primary analysis. Model second order relatives or future generations in a scenario analysis noting the sources of uncertainty.
	Choice of time horizon: a lifetime time horizon is usually recommended but when accounting for cascade effects, the individuals being considered may have differing life expectancies.	Adopting a time horizon based on the lifetime of the youngest individuals included in the study.
Costing	Identifying and quantifying health resource consumption: surveillance or treatment protocols initiated can vary from person to person (depending on age, phenotype, other risk factors) and different people may require different volumes of the same resources.	Base assumptions based on clinical practice guidelines or inputs from clinical experts and assess uncertainty in sensitivity analyses.
		Use a variety of data sources, including probands’ electronic medical records (EMRs), family members’ EMRs, and administrative health insurance databases, to capture individual-level data.
	It may be difficult to separate the costs of implementing a technology in an index case from the cost of cascade testing.	Collect and report disaggregated costs whenever possible.
Measurement and Valuation of Health Outcomes	QALYs cannot be aggregated across family members of different ages because different instruments and approaches are used to measure utilities in children and adults.	Conduct the analysis separately by age group, e.g., in children and adults, and report mean and incremental costs and health outcomes per person for each group, and in the combined cohort.
		Limit analysis to a pediatric or adult cohort only.
	QALYs cannot be aggregated across multiple individuals because they are defined and interpreted in terms of an individual’s life expectancy.	Report disaggregated outcomes as described above.
	Aggregating QALYs gives cascade QALYs equal weight to index patient QALYs and may unintentionally shift decisions toward benefiting family members over the patients because the QALY gains that multiple relatives experience may be greater on average than the QALY gains experienced by an individual patient.	Report disaggregated outcomes as described above.
		Report QALY gains separately and combined for index patients and relatives. The measurement of cascade QALYs must consider the prevalence of positive findings in relatives and the QALY calculation for relatives can include negative cases.
Model Design	Designing one model with health states and clinical events relevant to index patients and their relatives is complicated because probands and family members m ay have different clinical experiences and different probabilities of transitioning between health states.	Construct multiple decision analytic models, one for index cases and the others for family members with similar trajectories, and report costs and health outcomes both separately and combined.
	Including probands and family members as part of one cohort that “passes through” the model simultaneously is challenging.	Use advanced modeling techniques, such as discrete event microsimulations, which track the progress of individual persons with diverse attributes through health states.

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
