# Peer review of "Incorporating Cascade Effects of Genetic Testing in Economic Evaluation: A Scoping Review of Methodological Challenges"

_children, 2021, doi:10.3390/children8050346_

Round 1

Reviewer 1 Report

Cernat and colleagues have undertaken a scoping review to address their research question “How can methods for economic evaluation be modified to include cascade effects from genetic testing to improve the comprehensiveness and quality of evidence?”.  The manuscript is well-written and provides a broad overview of studies looking at cascade genetic testing in order to make recommendations to improve on methodological challenges they summarise from the studies.

I don’t have any major issues with the manuscript in its current form but only minor suggestions:

  1. Page 2, line 48: suggest changing “receive” to “undergo”
  2. Page 3, line 143: what does “hand-searching” mean?  Do the authors mean manual search?  Of what?
  3. Figure 2 - can the resolution of this figure be improved?  It’s quite blurry

Author Response

COMMENT

Cernat and colleagues have undertaken a scoping review to address their research question “How can methods for economic evaluation be modified to include cascade effects from genetic testing to improve the comprehensiveness and quality of evidence?”. The manuscript is well-written and provides a broad overview of studies looking at cascade genetic testing in order to make recommendations to improve on methodological challenges they summarise from the studies.

I don’t have any major issues with the manuscript in its current form but only minor suggestions:

Page 2, line 48: suggest changing “receive” to “undergo”

RESPONSE

“Receive” has been changed to “undergo” (page 2, line 60).

COMMENT

Page 3, line 143: what does “hand-searching” mean? Do the authors mean manual search? Of what?

RESPONSE

The description of the manual search in the methods (page 3, lines 128-130) has been edited as follows:

“The electronic search was supplemented with a manual search consisting of a review of the reference lists of included papers. Publications known to us that were not captured in the search were also included.”

The first paragraph of the results section has also been modified for clarify (page 4, lines 170-171):

“Four additional articles not captured in the electronic search were also included, one retrieved through the manual search of reference lists of included papers, and three that were previously known.”

COMMENT

Figure 2 – can the resolution of this figure be improved? It’s quite blurry.

RESPONSE

A new version of Figure 2 has been inserted and the image resolution settings for the document have been changed.

Reviewer 2 Report

This is a well written and relevant paper on the topic of incorporating cascade effects of genetic testing in economic evaluation. Although the authors provide reasoning for not undertaking a systematic review, I would expect a little more insight in existing literature/guidance on the topic (see also comments below).

I suggest to accept this manuscript with minor revisions.  

Specific comments/suggestions:

Abstract:

Line 13: “[…], and is particularly relevant for child health […]”

I would argue that this is not why it is “particularly” relevant to child health (it similarly relevant to adult health) and thus not an argument to publish in this journal. See also your own argumentation in line 61 “However, there are methodological challenges associated with incorporating cascade health service use in economic evaluation of technologies directed at children and adults”

Nevertheless, I would argue it is extra relevant to child health, because implications of cascading from (family members of) children could have particularly long-term impacts and is complicated by the fact that children are especially vulnerable and not always able to make complicated decisions. This makes (economic) evaluation and insight into the valuation of effects even more pertinent for this group. The authors could make this more explicit: consider to rephrase.

Line 26: “[…] as studies may currently be underestimating both the cost and health benefits attributable to the implementation of genetic technologies.”

It is not clear to the reader whether this current underestimation is a conclusion of this study (as I would expect from the last sentence of the abstract) or a hypothesis. Perhaps rephrase and start the sentence with this part: “As studies may currently […], we conclude that development of methods to address current difficulties is required.”

Introduction:

Line 31: “Genetic testing is a powerful diagnostic tool that can be applied to (suspected) hereditary diseases affecting any organ in the body.”

Consider to insert hereditary to describe the correct context: it will not be useful for any disease. Furthermore I would suggest to add a sentence hereafter to shortly describe the direct value of a genetic diagnosis for the index patient (often the “main” value).

Line 43: “There are also costs to the health care system: physicians fees for follow-up appointments […]”

Consider changing “follow-up appointments” to “pre- and post-test counseling and follow-up appointments”.

Line 72: “Studies may have been conducted with one of two aims”

Consider rephrasing to “to assess the overall impact of implementing a genetic technology for index case identification”, to make clear that this includes not only impact on the index.

Line 78: “The guiding research question was: “How can methods for economic evaluation be modified to include cascade effects from genetic testing to improve the comprehensiveness and quality of evidence?””

You do not seem to describe currently existing guidance/models/methods that include these effects, and in in your methods exclude existing reviews on the topic (e.g. When is Genomic Testing Cost-Effective? Testing for Lynch Syndrome in Patients with Newly-Diagnosed Colorectal Cancer and Their Relatives (nih.gov)). Given the purpose of the review “to provide an overview of the literature and to advance economic evaluation methods”, I would expect that a short overview of (a lack of) existing guidance is given at least to substantiate the reasoning behind this research question. See also:

Materials and methods:

Line 117: “In-eligible studies were […], or qualitative, […] case reports […], editorials, commentaries […]”

Since no overview of (a lack of) existing guidance is given in the introduction, I would have expected that these types of studies that addressed the topic would have been included or at least used as a starting point to select studies that were included in these reports (this is not the case for the Grosse paper mentioned in the previous comment). Otherwise it seems like the authors are “re-inventing the wheel”.  

Line 188: “ […] were compared to children were not tested who initiated statin treatment […]

Check grammar/consider rephrasing

Line 227: “This paper differs from others included in this review in that cascade genetic testing was not conducted for the benefit of the probands’ relatives, but rather to determine inheritance of a variant identified in the index case.”

Since in this case (healthy) parents were sequenced to identify whether the VUS at hand is a de novo (and thus more likely pathogenic) variant, this is in my opinion not considered as cascade testing. In general I think the sequencing of a trio (a patients and his/her parents) is considered as having no direct consequences for the parents (it does not inform them of their own risk) and thus is part of the (costs of) the diagnostic test. Therefore consider excluding this study. If the authors however consider the costs of testing the parents are the “cascade effects” this should be mentioned more explicitly.

Table 1:

Please check for consistency of wording and/or explain what is meant with e.g. health system/health care system perspective (seems most inconsistent/unclear for the “perspective and time horizon” column).

Discussion:

Line 361: “There are a number of methodological challenges common to economic […]”

The identification of the challenges seems to be part of the results of this study: please consider putting them in the results section (together with a short description of the challenges). Furthermore: please elaborate on how you came to this specific set of challenges.

The suggested solutions to the different challenges is well described and the provided overview in table 2 is of added value to the reader. Furthermore the conclusion is clear and concise, although I would have liked the authors to elaborate a little more on what they think is needed (who needs to do what?) for the suggested development and validation of formal methods. This could increase the impact of the publication. 

Author Response

COMMENT

This is a well written and relevant paper on the topic of incorporating cascade effects of genetic testing in economic evaluation. Although the authors provide reasoning for not undertaking a systematic review, I would expect a little more insight in existing literature/guidance on the topic (see also comments below). I suggest to accept this manuscript with minor revisions. Specific comments/suggestions:

Abstract:

Line 13: “[…], and is particularly relevant for child health […]”

I would argue that this is not why it is “particularly” relevant to child health (it similarly relevant to adult health) and thus not an argument to publish in this journal. See also your own argumentation in line 61 “However, there are methodological challenges associated with incorporating cascade health service use in economic evaluation of technologies directed at children and adults”

Nevertheless, I would argue it is extra relevant to child health, because implications of cascading from (family members of) children could have particularly long-term impacts and is complicated by the fact that children are especially vulnerable and not always able to make complicated decisions. This makes (economic) evaluation and insight into the valuation of effects even more pertinent for this group. The authors could make this more explicit: consider to rephrase.

RESPONSE

The first sentence of the abstract has been modified (page 1, lines 14-15):

“Cascade genetic testing is indicated for family members of individuals testing positive on a genetic test, and is particularly relevant for child health because of their vulnerability and the long-term health and economic implications.”

The following sentence has been added to the Introduction (page 2, lines 80-81):

“These difficulties are exacerbated when both children and adults are considered in an economic evaluation simultaneously.”

COMMENT

Line 26: “[…] as studies may currently be underestimating both the cost and health benefits attributable to the implementation of genetic technologies.”

It is not clear to the reader whether this current underestimation is a conclusion of this study (as I would expect from the last sentence of the abstract) or a hypothesis. Perhaps rephrase and start the sentence with this part: “As studies may currently […], we conclude that development of methods to address current difficulties is required.”

RESPONSE

The last sentence of the abstract has been revised as you suggest (page 1, lines 25-28):

“As health economic studies may currently be underestimating both the cost and health benefits attributable to genetic technologies through omission of cascade effects, development of methods to address these difficulties is required.”

COMMENT

Introduction:

Line 31: “Genetic testing is a powerful diagnostic tool that can be applied to (suspected) hereditary diseases affecting any organ in the body.”

Consider to insert hereditary to describe the correct context: it will not be useful for any disease. Furthermore I would suggest to add a sentence hereafter to shortly describe the direct value of a genetic diagnosis for the index patient (often the “main” value).

RESPONSE

The beginning of the paper has been modified as follows:

“Genetic testing is a powerful diagnostic tool that can be applied to diseases affecting any organ system in the body [1-4]. These diseases may be hereditary or mutations can arise de novo. Identification of a genetic diagnosis in a patient facilitates more appropriate management and enables physicians to better approximate patients’ prognoses. Importantly, identifying a genetic diagnosis in a patient enables genetic testing or clinical screening of their family members to determine whether they may also be at risk of developing the disease [3-5].”

Reference #1

Dillon, J.O., Lunke, S., Stark, Z., Yeung, A., Thorne, N., Melbourne Genomics Health Alliance, Gaff, G., White, S.M., Tan, T.Y. Exome sequencing has higher diagnostic yield compared to simulated disease-specific panels in children with suspected monogenic disorders. Eur. J. Hum. Genet. 2018, 26, 644–651.

Reference #2

Boland, P.M., Yurgelun, M.B., Boland, C.R. Recent progress in lynch syndrome and other familial colorectal cancer syndromes. CA. Cancer J. Clin. 2018, 68(3), 217-231.

Reference #3

Ahluwalia, M., Ho, C.Y. Cardiovascular genetics: The role of genetic testing in diagnosis and management of patients with hypertrophic cardiomyopathy. Heart. 2021, 107(3), 183-189.

Reference #4

Kalsner, L., Twachtman-Bassett, J., Tokarski, K., Stanley, C., Dumont-Mathieu, T., Cotney, J., Chamberlain, S. Genetic testing including targeted gene panel in a diverse clinical population of children with autism spectrum disorder: Findings and implications. Mol. Genet. Genomic Med. 2018, 6(2), 171-185.

Reference #5

Knowles, J.W., Rader, D.J., Khoury, M.J. Cascade screening for familial hypercholesterolemia and the use of genetic testing. JAMA. 2018, 318(4), 381-382.

COMMENT

Line 43: “There are also costs to the health care system: physicians fees for follow-up appointments […]”

Consider changing “follow-up appointments” to “pre- and post-test counseling and follow-up appointments”.

RESPONSE

The sentence on page 2, line 55 has been changed as follows:

“There are also costs to the health care system: physician and genetic counseling fees for pre- and post-test counseling and follow-up appointments…”

COMMENT

Line 72: “Studies may have been conducted with one of two aims”

Consider rephrasing to “to assess the overall impact of implementing a genetic technology for index case identification”, to make clear that this includes not only impact on the index.

RESPONSE

The sentence has been modified as you suggest (page 2, line 89):

“Studies may have been conducted with one of two aims: either to assess the overall impact of implementing a genetic technology for index case identification, or to evaluate only the cascade that ensues.”

COMMENT

Line 78: “The guiding research question was: “How can methods for economic evaluation be modified to include cascade effects from genetic testing to improve the comprehensiveness and quality of evidence?””

You do not seem to describe currently existing guidance/models/methods that include these effects, and in in your methods exclude existing reviews on the topic (e.g. When is Genomic Testing Cost-Effective? Testing for Lynch Syndrome in Patients with Newly-Diagnosed Colorectal Cancer and Their Relatives (nih.gov)). Given the purpose of the review “to provide an overview of the literature and to advance economic evaluation methods”, I would expect that a short overview of (a lack of) existing guidance is given at least to substantiate the reasoning behind this research question. See also:

Materials and methods:

Line 117: “In-eligible studies were […], or qualitative, […] case reports […], editorials, commentaries […]”

Since no overview of (a lack of) existing guidance is given in the introduction, I would have expected that these types of studies that addressed the topic would have been included or at least used as a starting point to select studies that were included in these reports (this is not the case for the Grosse paper mentioned in the previous comment). Otherwise it seems like the authors are “re-inventing the wheel”.

RESPONSE

In Canada and elsewhere, economic evaluation requirements are dictated by governmental health technology assessment agencies. It is the omission of guidance on incorporating cascade effects in these official guidelines that is of concern and motivated the present review. This has been clarified with following revisions:

The following sentence in the Introduction (page 1, line 63)

“Traditional economic evaluations only account for the impact of a new technology on the patient and cascade effects are not routinely considered [10]”

has been replaced with the following on page 1, lines 66-68:

“However, methods for economic evaluation stipulated by health technology assessment agencies in Canada, the United States and the United Kingdom fail to account for the cascade effects of a new technology [11-13].”

Reference #11:

CADTH. Guidelines for the economic evaluation of health technologies: Canada. Ottawa; 2017. 

New Reference #12:

National Institute for Health and Care Excellence (NICE). Guide to the methods of technology appraisal 2013. 2013.

New Reference #13:

Neumann, P.J., Sanders, G.D., Russell, L.B., Siegel, J. E., Ganiats, T.G., editors. Cost-effectiveness in health and medicine. 2nd ed. New York: Oxford University Press; 2017

As this review aimed to summarize the various approaches taken in actual studies of cascade effects, reviews and commentaries were excluded. However, the following revision has been made to the Introduction, page 2, lines 63-66 to elaborate on expert commentary:

“A past review has examined cost-effectiveness analyses (CEAs) for Lynch syndrome that incorporate cascade effects, and highlighted how changing the number of relatives included in analysis can modify the results of an economic evaluation [10].”

New Reference #10:

Grosse, S.D. When is genomic testing cost-effective? Testing for Lynch syndrome in patients with newly-diagnosed colorectal cancer and their relatives. Healthcare. 2015, 3, 860-868.

COMMENT

Line 188: “ […] were compared to children were not tested who initiated statin treatment […]

Check grammar/consider rephrasing

RESPONSE

The sentence has been modified (page 5, lines 219-220):

“Children who underwent genetic testing for FH and who began statin treatment after receiving a genetic diagnosis were compared to children who did not undergo genetic testing, but who initiated statin treatment either when they reached 25 years of age, or earlier if they experienced a cardiovascular event.”

COMMENT

Line 227: “This paper differs from others included in this review in that cascade genetic testing was not conducted for the benefit of the probands’ relatives, but rather to determine inheritance of a variant identified in the index case.”

Since in this case (healthy) parents were sequenced to identify whether the VUS at hand is a de novo (and thus more likely pathogenic) variant, this is in my opinion not considered as cascade testing. In general I think the sequencing of a trio (a patients and his/her parents) is considered as having no direct consequences for the parents (it does not inform them of their own risk) and thus is part of the (costs of) the diagnostic test. Therefore consider excluding this study. If the authors however consider the costs of testing the parents are the “cascade effects” this should be mentioned more explicitly.

RESPONSE

The following revision was made on page 6, lines 262-267:

However, studying the origin of a variant, and whether it is de novo or inherited, is routine in medical genetics and the costs of these confirmatory genetic tests are therefore relevant for economic evaluation of genetic testing technologies directed at probands. Although parents may be asymptomatic in these investigations, a positive finding may precipitate ongoing surveillance. It is for these reasons that this paper was included.

COMMENT

Table 1:

Please check for consistency of wording and/or explain what is meant with e.g. health system/health care system perspective (seems most inconsistent/unclear for the “perspective and time horizon” column).

RESPONSE

We have revised the table for consistency. There are now only three perspectives studies may have adopted: the public health care system perspective, the public payer perspective, and the societal perspective.

We were not attempting to draw any distinctions by use of health system vs health care system, and all instances of “health system” have now been changed to read “health care system.” Our decision to describe the perspective as that of the health care payer or that of the health care system was based on the authors’ own phrasing in each study (i.e., if the authors stated it was a health system perspective, we did the same; if they stated it was a health payer perspective, we did the same).

COMMENT

Discussion:

Line 361: “There are a number of methodological challenges common to economic […]”

The identification of the challenges seems to be part of the results of this study: please consider putting them in the results section (together with a short description of the challenges). Furthermore: please elaborate on how you came to this specific set of challenges.

RESPONSE

As a review paper, we felt that elaborating on the identification and implications of methodological challenges is best suited to the Discussion. In our Results section, we present how the included studies were conducted and describe in detail their methods and findings. Our Discussion is more interpretive in nature, as the challenges identified were not necessarily made explicit in the included studies. Thus we use the Discussion to consider what difficulties the methods of the included studies either exemplify or help ameliorate.

We have modified the first sentence in subsection 4.1 (page 17, line 413) to read:

“Based on close examination of included studies, there are a number of methodological challenges common to economic evaluations of cascade effects related to…”

COMMENT

The suggested solutions to the different challenges is well described and the provided overview in table 2 is of added value to the reader. Furthermore the conclusion is clear and concise, although I would have liked the authors to elaborate a little more on what they think is needed (who needs to do what?) for the suggested development and validation of formal methods. This could increase the impact of the publication.

RESPONSE

The last sentences of the Conclusion have been revised as follows:

“Development and validation of formal methods by health economists and health services researchers to address these challenges is warranted. Further, in future updates of their guidelines, HTA agencies around the world must explicitly consider cascade effects and optimal approaches for incorporation in economic evaluation.”